# Dental Implant Treatment in Patients Suffering from Oral Lichen Planus: A Narrative Review

**DOI:** 10.3390/ijerph19148397

**Published:** 2022-07-09

**Authors:** Bartłomiej Górski

**Affiliations:** Department of Periodontology and Oral Mucosa Diseases, Medical University of Warsaw, Binieckiego 6 St., 02-097 Warsaw, Poland; bgorski@wum.edu.pl; Tel.: +48-22-270-16-16

**Keywords:** complications, dental implants, implant survival, oral lichen planus

## Abstract

Background: The aim of this study was to describe the complications and survival rates of dental implants placed in patients suffering from oral lichen planus (OLP) and to present recommendations for implant treatment in this group of patients through a narrative review of the published studies. Methods: A search of the literature was conducted using four databases: PubMed/Medline, Web of Science, Cochrane, and Scopus with a stop date of May 2022. Results: Eighteen studies were evaluated. The results showed that dental implant survival rates in patients with OLP were similar to those reported in the general population. Moreover, the existing literature seemed to imply that OLP is not a suspected risk factor for peri-implant diseases. However, patients suffering from erosive forms of OLP or desquamative gingivitis and poor oral hygiene were more susceptible to developing peri-implant diseases; in addition, oral squamous cell carcinoma was observed in a few cases of OLP. Conclusion: With the limitations of this narrative review, dental implants may be regarded as a safe and feasible therapeutic approach to the treatment of patients with well-controlled OLP. These patients should be monitored carefully during follow-up care. Well-designed prospective trials are required to validate the present findings.

## 1. Introduction

Oral lichen planus (OLP) is a chronic inflammatory disorder that affects the mucous membranes with clinical outbreaks and with periods of remission [1]. It is more frequent in women aged between 40 and 60. Its prevalence worldwide was reported to be 1.01% in the adult population, reaching 1.43% in Europe [2]. The etiology of OLP is still ambiguous, however, it involves cell-mediated immune dysregulation brought about by the interaction between genetic and environmental factors. The latter can be divided into local factors (metal dental restorations, trauma, cigarette smoking, alcohol intake), systematic diseases (diabetes mellitus, hypertension, anxiety), and the consumption of certain drugs (antimalarials, antihypertensive, diuretics, non-steroidal anti-inflammatory drugs) [1]. The oral lesions can occur in six clinical forms: reticular, plaque-like, papular, atrophic/erosive, ulcerative, and bullous types (Figure 1). Erosive and atrophic OLP is mostly symptomatic and pain can be one of its main characteristics. OLP is repeatedly associated with the emergence of new lesions in the healthy mucosa following an injury or trauma (Köbner phenomenon) [3]. Moreover, in almost half of the patients, gingival involvement may be present which is characterized by desquamation, bleeding, pain, erosions, and ulcerations [4] (Figure 2). It should also be noted that OLP is a potentially malignant disorder with an overall transformation rate of 1.40% [5].

In healthy patients, fixed and removable implant-supported restorations exhibited high implant survival rates and acceptable bone loss with considerable improvement in the quality of life and satisfaction of patients. Even though the reported long-term success and implant survival rate was above 90%, a very high proportion (8–44%) of dental implants developed peri-implant mucositis or peri-implantitis [6,7,8]. Peri-implant diseases may be associated with certain risk factors: poor oral hygiene, smoking, history of periodontitis, and systematic diseases such as diabetes. A single factor alone may not influence the risk measurably, whereas a combination of multiple factors may have a significant impact. It was also suggested that OLP may negatively affect the attachment of the mucosal epithelium to prosthetic surfaces in patients treated with dental implants [9]. Furthermore, a local increase in pro-inflammatory cytokine levels and changes in the expression of molecules responsible for cell adhesion may occur [10]. For this reason, the eligibility of patients suffering from OLP to receive implant placement was questioned [11]. In recent years, the spectrum of indications for dental implant placement has widened. Although the literature in this field is scant and no international guidelines exist with respect to dental implant treatment in patients with muco-cutaneous autoimmune diseases, this therapeutic approach is currently preferred in OLP patients. Nevertheless, dental implant placement in patients suffering from OLP should be considered with regard to potential risks and complications that may impinge on the outcomes.

The aim of this study was to describe the complications and survival rates of dental implants placed in patients suffering from OLP through a narrative review of the published literature. An additional goal was to present evidence-based recommendations for implant treatment in this group of patients.

## 2. Materials and Methods

A search of the literature was performed in June 2022 using four databases: PubMed/Medline, Web of Science, Cochrane, and Scopus. All randomized clinical trials (RCT), cohort studies, case-control studies, case series and case reports, systematic reviews, and meta-analyses on dental implant treatment in humans with oral lichen planus until May 2022 were included. Animal studies, in vitro studies, abstracts, and narrative reviews were excluded. All selected articles were in English.

## 3. Results

One hundred and eighty-one studies were identified after the initial screening of titles and abstracts. Subsequently, duplicates, abstracts, studies of the extraoral localization of lesions, animal studies, in vitro studies, and manuscripts published in languages other than English were excluded. After full-text reading, eighteen studies (ten case reports, two case series, one case-control retrospective, one case-control prospective, one cross-sectional, and three retrospective cohort studies) reporting on implant treatment in patients with OLP were evaluated [12,13,14,15,16,17,18,19,20,21,22,23,24,25,26,27,28,29]. A list of these studies is provided in Table 1.

In 2000, Esposito et al. [12] published a case report on the loss of implants where, apart from erosive OLP, parafunction and poor bone quality were diagnosed. According to the authors, no correlation between implant failure and OLP could be established. Three years later another case report on two patients with successful outcomes was published [22]. Then, three other favorable cases of implant-supported fixed prostheses and two cases of implant-retained overdenture in OLP were reported [13,20,21,23]. 

Czerninski et al. [24] compared OLP patients treated with dental implants (the study group) to those who had not (control group). No implant failures were recorded at follow-up (range 12–24 months), nor did implant placement influence the disease manifestations. The implant survival rate was identical to that of non-OLP edentulous patients.

Only three studies compared OLP patients with healthy controls [25,26,28]. The first controlled prospective study including 18 patients with OLP was published in 2012. Hernández et al. [25] reported an implant survival rate of 100% for the OLP group. Quite surprisingly, a lower rate of survival (96.8%) was found in the control group; however, the difference was not statistically significant. Peri-implant mucositis was detected in 44.6% of the implants and 66.6% of the patients with OLP. The presence of desquamative gingivitis was associated with a higher rate of peri-implant mucositis for implants in the OLP group (*p* = 0.004). Peri-implantitis appeared in 10.7% of the implants and 27.7% of the patients with OLP (*p* = ns). López-Jornet et al. [26] found no differences between OLP patients and the control group with respect to implant survival, peri-implant mucositis, peri-implantitis, and marginal bone loss. The overall success rate in the OLP and control groups was 96.42% and 92%, respectively. Peri-implant mucositis and peri-implantitis were detected in 17.86% and 25% of the OLP group, whereas the control group showed 18% and 16%. Peri-implantitis was more frequent in the mandible and the posteriorly placed dental implants. These results also suggested that implants did not influence the manifestation of OLP and that OLP was not a risk factor for peri-implant diseases. Khamis et al. [28] divided patients into 3 groups and studied them over 4 years: healthy individuals, OLP patients controlled using low doses of systemic corticosteroids, and noncontrolled OLP patients. There was no statistically significant difference in marginal bone loss between healthy and controlled OLP patients; however, noncontrolled OLP patients exhibited increased marginal bone loss (*p* < 0.001), which reached 2.53 mm. Histopathologic examination of analyzed biopsies showed a healthy tissue architecture in the controlled patients, whereas inflammatory cellular infiltration was observed for the non-controlled patients. 

Aboushelib and Elsafi [27] recommended administering oral corticosteroids and soft tissue laser irradiation before insertion of dental implants in patients with OLP. They reported an overall success rate of 24% for the uncontrolled patients; as for the 55 inserted implants, 42 failed after a short time (7 to 11 weeks). On the other hand, the success rate in the controlled group was 100% after 3 years of clinical observation. Those patients were put on an ascending dose (5 mg/10 days) of oral corticosteroids until a daily dose of 20 mg/day was achieved and maintained for 2 weeks. Anitua et al. [29] administrated deflazacort 30 mg starting two days prior to placement of short dental implants (≤8.5 mm) without bone augmentation, then 15 mg postoperatively for three days and 7.5 mg for another three days. All surgeries were performed outside the flare-up periods of OLP, and a prophylactic regimen of oral corticosteroids was administered to avoid flare-ups after the procedure. Sixty-five of the 66 implants survived with a mean follow-up of 68 months, and there were no significant differences between erosive and reticular OLP. The success rate was 98%. 

The occurrence of oral squamous cell carcinoma around titanium dental implants in patients with OLP was reported in six clinical reports and with respect to nine patients [14,15,16,17,18,19]. The implant failure rate in these cases was high, but none of the implants lost osseointegration, instead, they were removed together with tumors. Those cases may present a process of malignant conversion from OLP to oral squamous cell carcinoma via epithelial hyperplasia, although the malignant transformation rate was low.

## 4. Discussion

This review was focused on the outcomes of dental implants in patients with OLP. Implant survival rates in patients with OLP were comparable to those reported in the general population, thus this treatment approach may be considered a safe solution for this category of patients. The loss of implants in some cases was not due to OLP but rather to parafunction, poor bone quality, or malignant transformation of OLP [3]. Dental implant placement did not negatively influence the course of OLP, nor was it a risk factor for peri-implant diseases [24,26]. However, patients suffering from erosive oral lichen planus who additionally had desquamative gingivitis were more likely to develop peri-implant mucositis and they exhibited a higher rate of peri-implantitis compared to non-desquamative gingivitis OLP subjects [25,26,28]. The prevalence of peri-implantitis ranging from 10.7% to 25% was only reported in two studies [25,26]. Those discrepancies may be explained by differences in the definitions of peri-implantitis, varied prosthetic designs, or maintenance programs. A closer look should also be taken at the study by Aboushelib et al. [27]. When the first set of 55 dental implants was placed in the presence of active lesions without prior medication, the survival rate was only 23.6%. However, for the second set of 42 implants, which were subsequently placed after oral corticoid administration and soft tissue laser irradiation, the survival rate was 100% (36 months of follow-up). When the results of the abovementioned study and of the cases that developed oral squamous cell carcinoma were not considered, the dental implant failure rate in OLP patients was as low as 2.7% after a follow-up of five years, which is similar to populations without OLP (~2%) [30]. Recent systematic reviews reported a survival rate of dental implants placed in OLP patients of 98.9% after 38 months, 97.3% after 5 years, or 95.8% during a follow-up period ranging from 1 to 13 years [20,30,31]. The results from other systematic reviews and meta-analyses evaluating the outcomes of dental implants in OLP patients are provided in Table 2 [20,30,31,32,33,34,35,36,37].

Conventional prosthetic treatment options and improvements in patients’ quality of life following therapy might be limited due to the effects of the underlying disease. In a very recent systematic review, Anitua et al. [31] made some clinical recommendations for implant-prosthetic treatment in patients with OLP. Fixed implant-supported prostheses may prevent the friction or trauma caused by conventional removable mucogingival prostheses or overdentures. All corners and edges of restorations in contact with the buccal mucosa or the margin of the tongue need to be smoothed or rounded. The design of restorations must create proper access for performing plaque control to reduce peri-implant mucositis and peri-implantitis. Screw-retained restorations are more suitable than cemented ones due to their easier retrievability during obligatory and regular check-ups of the underlying mucosa. The connection system between implant abutments should be as precise as possible. The use of transmucosal abutments may allow an epithelium to face the most biocompatible materials with optimal superficial characteristics. Regarding the most suitable materials for OLP patients, the choice of titanium for the abutments and the metal suprastructure of restorations may be recommended [31]. No data were reported on the usage of zirconia (ZrO_2_) abutments in this systematic review. However, many in vitro and in vivo studies revealed clear advantages for zirconia implant-prosthetic components with regard to the interaction with soft tissue cells such as fibroblasts, blood cells, and epithelial cells, as well as protein adsorption, cell alignment, and biocompatibility [38]. Advantages in relation to periodontal seals were demonstrated by better attachment and alignment of collagen fibers with components developed in zirconia compared with other recognized dental materials. Surface treatments of zirconia showed excellent osseointegration and provided encouraging prospects for rapid bone adhesion. Moreover, less bacterial adhesion on smoother zirconia ceramic components, mainly prosthetic components, was demonstrated. Blood flow in the tissue surrounding zirconia abutments was similar to that in the soft tissue around natural teeth, which could be advantageous for the maintenance of immune function by improving blood circulation [39]. In another study, lithium disilicate showed comparable biological properties to titanium alloy as an implant abutment material [40]. Ceramics should be preferred over resins for definitive treatments due to a decreased accumulation of biofilm to feldspathic ceramics. Metals such as gold, mercury, palladium, nickel, and copper could be associated with OLP, especially with oral lichenoid lesions, and thus should be avoided [41]. 

It is commonly accepted that implant placement should be carried out during phases of remission of OLP. There are no broadly approved treatment guidelines for active OLP, however, the treatment of choice is topically or systematically administered steroids [42]. The beneficial effect of low doses of oral corticosteroids on dental implant maintenance and the reduction in the recurrence of OLP was also reported [27,28,42]. In a very recent systematic review by Torrejon-Moya et al. [36], the authors recommended prophylactic corticosteroid therapy in order to reactivate erosive and atrophic OLP after implant placement. Deflazacort 30 mg 2 days pre-operatively, 15 mg 3 days post-operatively, and 7.5 mg for a further 3 days was proposed, together with mouthwashes of 0.01% triamcinolone acetonide three times per day until the remission of the acute forms. On the other hand, systematic corticosteroid treatment in OLP patients was associated with decreased bone mineral density, especially during the first six months of therapy [43]. The thorough maintenance of oral hygiene and patients’ compliance appeared to be crucial factors for uneventful dental implant therapy. Meticulous plaque control and regular follow-ups are important to control peri-implant mucositis and peri-implantitis, as well as for the early detection of malignant transformation in the vicinity of dental implants. OLP lesion and oral squamous cell carcinoma could be confused with peri-implantitis and for this reason, a prompt biopsy should be performed. 

There are some crucial aspects that should be considered when interpreting available scientific data on dental implant treatment in patients with OLP. Most of the studies were retrospective and nonrandomized, and there were a large number of case reports and small case series, which could not provide the full information regarding the implantation procedure and possible risk of bias. More robust evidence could be gained from randomized clinical trials and multicenter studies. There is a need for large, prospective, and well-designed research in this area. Another important issue is the fact that only a few studies implemented biopsies to diagnose OLP. However, both clinical and histopathologic criteria need to be present to make the correct diagnosis. Furthermore, case definitions for peri-implant diseases varied considerably in the available studies. Last but not least, the disadvantages that are adherent to the design of this narrative review (higher degree of bias) should be considered before drawing a conclusion [44].

## 5. Conclusions

With the limitations of this narrative review, dental implants may be regarded as a viable therapeutic approach in the treatment of patients suffering from oral lichen planus. Periods of implant survival seem to be comparable to those of patients without OLP, and implant placement did not influence OLP manifestations. Implant-prosthetic treatment guidelines regarding healthy patients should be strictly followed. No patient should be treated during a flare-up period of the disease and meticulous oral hygiene and regular appointments are important to prevent inflammatory tissue response. 

## Figures and Tables

**Figure 1 ijerph-19-08397-f001:**
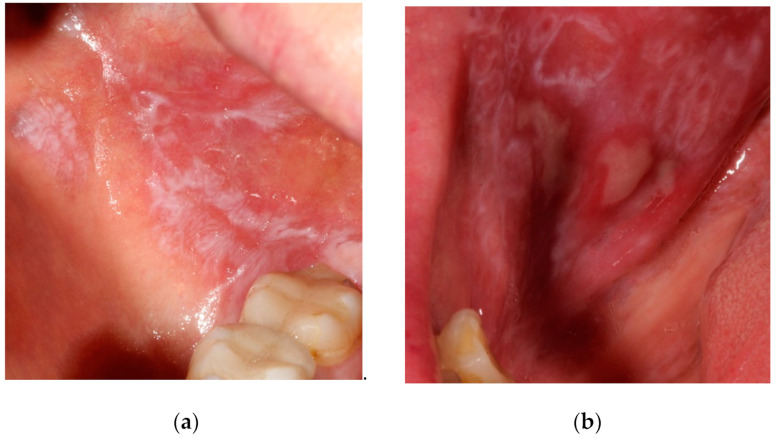
Lesions on buccal mucosa in patients with oral lichen planus (**a**) reticular type; (**b**) erosive part.

**Figure 2 ijerph-19-08397-f002:**
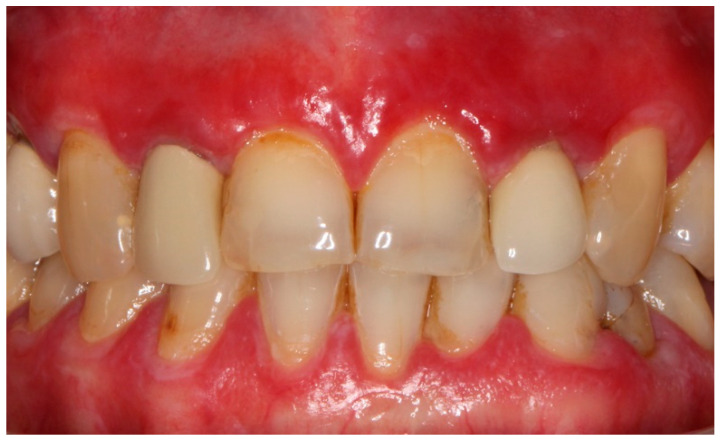
Desquamative gingivitis in a patient with oral lichen planus.

**Table 1 ijerph-19-08397-t001:** Main characteristics of the selected studies and summary of reported outcomes.

Authors	Study Design	Patients	Implants (Number, Brand)	Control	Follow-Up Time (Months)	Implant Survival Rate (%)	Complications
Esposito et al. [12]	Cr	1 female (erosive OLP), 69 y.	2 Brånemark implants	None	32, 60	0	Implant failure in a patient with parafunction and poor bone quality.
Oczakir et al. [13]	Cr	1 female with OLP, 74 y.	4 implants (brand not reported)	None	72	100	No complications
Czerninski et al. [14]	Cr	1 female (erosive OLP), 52 y.	3 implants (brand not reported)	None	36	-	Oral squamous cell carcinoma developed around dental implants in a heavy smoker patient
Gallego et al. [15]	Cr	1 female (reticular OLP), 81 y.	2 implants (brand not reported)	None	36	0	Implant loss was caused by partial mandibular resection due to oral squamous cell carcinoma developed around one implant.
Marini et al. [16]	Cr	1 female with plaque-type OLP, 51 y.	2 implants (brand not reported)	None	108	50	Post-treatment evolution of OLP to oral squamous cell carcinoma with loss of implant
Moergel et al. [17]	Cr	3 females with OLP, 54/69/80 y.	The number and the brand of implants not reported	None	6-51	-	Oral squamous cell carcinomas developed around dental implants, one patient had a history of cancer and the other two were smokers
Raiser et al. [18]	Cr	2 females with OLP, 55/70 y.	10 implants (brand not reported)	None	96.3	100	Oral squamous cell carcinoma around dental implants
Noguchi et al. [19]	Cr	1 female with OLP, 78 y.	4 implants (brand not reported)	None	48	0	Post-treatment evolution of OLP to oral squamous cell carcinoma with loss of implants
Fu et al. [20]	Cr	1 female with erosive OLP, 65 y.	4 NB implants	None	36	100	No complications
Martin-Cabezas [21]	Cr	1 female with erosive OLP, 83 y.	3 implants (brand not reported)	None	12	100	Peri-implantitis
Esposito et al. [22]	CS	2 females (erosive OLP), 72/78 y.	4 Straumann implants	None	21	100	No complications
Reichert et al. [23]	CS	3 females (Pt I: reticular OLP, Pt II: reticular and atrophic OLP, Pt III: atrophic OLP without erosions), 63/68/79 y.	8 implants (2 HATI, 1 ZL Microdent, 5 not reported)	None	Reported only for one Pt: 36	100	Pt I: delayed wound healing; Pt II: bone resorption and gingivitis; Pt III: no complications
Czerninski et al. [24]	CCR	14 patients: 11 females, 3 males (reticular, erosive and atrophic OLP), mean age 59.5 y.	54 implants (brand not reported)	15 controls: 11 females, 4 males with OLP, mean age 59.1 y., without dental implants	12–24	100 in both groups	Bleeding on probing and gingivitis: nine implants in three patients
Hernández et al. [25]	CCP	18 patients: 14 females and 4 males (erosive OLP), mean age 53.7 y.	56 NB Ti-Unite implants	18 controls: 12 females and 6 males without OLP, mean age 52.2 y., 62 implants	53.5 (tests) 52.3 (controls)	100 (tests) 96.77 (controls)	Peri-implant mucositis: 12 (43%) patients with OLP and 16 (57%) patients without OLP. Peri-implantitis: 5 (55.6%) patients with OLP and 4 (44.4%) patients without OLP. Two implants failed in the control group 32 and 46 months after loading.
López-Jornet et al. [26]	CCCS	Group I: 16 patients: 10 females, 6 males (11 reticular OLP, 5 atrophic erosive OLP), mean age 64.5 y.	56 implants (brand not reported)	Group II: 16 controls: 11 females, 5 males (9 reticular OLP, 6 atrophic-erosive OLP), mean age 63 y., without dental implants. Group III: 16 controls: 8 females, 8 males without OLP, mean age 42, with 50 implants	42 (12–120)	96.4 (Group I) 92 (Group III)	Peri-implant mucositis: 17.8% in the OLP-implant group and 18% in the control group. Peri-implantitis: 25% in the OLP-implant group and 16% in the control group. Two mobile implants were found in Group I, and four mobile implants in Group III.
Aboushelib et al. [27]	CP	23 patients: 12 females, 11 males with active OLP, mean age 56.7 y.	First set: 55 Zimmer implants Second set: 42 Zimmer implants (+oral corticosteroids and low-energy soft tissue laser irradiation at the implant insertion)	None	3 36	23.6 100	No osseointegration No complications
Khamis et al. [28]	CR	20 patients with controlled OLP (by administration of low dose of corticoids)	The number and brand of implants not reported	49 controls: 17 subjects without OLP without dental implants. 22 subjects with noncontrolled OLP with dental implants	48	100	Non controlled patients with OLP with dental implants exhibited increased marginal bone loss (up to 2.53 mm after 4 years) and recurrence of the oral lesions
Anitua et al. [29]	SCR	23 patients: 20 females, 3 males (15 reticular OLP, 8 erosive OLP), mean age 58 y.	66 BTI implants	None	68	98.4	Implant removal due to recurrent gingivitis in one patient

Cr: case report; CS: case series; OLP: oral lichen planus; y: years; Pt: patient; CCR: case-control retrospective; CCP: case-control prospective; CCCS: case-control cross-sectional; CP: cohort prospective; CR: cohort retrospective; SCR: single cohort retrospective.

**Table 2 ijerph-19-08397-t002:** Main outcomes of systematic reviews analyzing dental implant treatment in patients with oral lichen planus.

Authors	Year	Study Design	Type of Included Studies, Number of Patients, and Number of Implants	Main Outcomes
Fu et al. [20]	2019	Systematic review	13 studies (9 case reports, 1 case-control prospective study, 1 case-control cross-sectional study, 1 case-control retrospective study, 1 cohort retrospective study) with 86 patients and 259 implants	“The survival rate of implants was 95.8% during a follow-up period ranging from 1 to 13 years. Dental implants seem to be an acceptable and reliable treatment option in patients with OLP.”
Chrcanovic et al. [30]	2020	Systematic review	22 studies (15 case reports, 1 case-control retrospective study, 1 case-control cross-sectional study, 1 case-control prospective study, 4 cohort retrospective studies) with 230 patients and 615 implants	“The overall implant failure rate was 13.9% (85/610). In patients with oral squamous cell carcinoma (OSCC) the failure rate was 90.6% (29/32), but none of these implants lost osseointegration; instead, the implants were removed together with the tumor. One study (Aboushelib et al. 2017) presented a very high implant failure rate, 76.4% (42/55), in patients with “active lichen planus”, with all implants failing between 7–16 weeks after implant placement (…). If OSCC patients and the cases of the latter study are not considered, then the failure rate becomes very low (2.7%, 14/523). The time between implant placement and failure was 25.4 ± 32.6 months (range 1–112).”
Anitua et al. [31]	2021	Systematic review	8 studies (2 case series, 1 case-control retrospcetive study, a case-control cross-sectinal study, 1 case-control prospective study, 3 cohort restrospective studies) that involved 141 patients and 341 implants	“The weighted mean follow-up was 38 months and the weighted mean survival of the implants 98.9%. No statistical differences were observed between cemented or screw retained prostheses and the materials employed or the technology to manufacture the prostheses.”
Reichart et al. [32]	2016	Systematic review	9 studies (6 case reports, 1 case-control retrospective study, 1 case-control cross-sectional study, 1 case-control prospective study)	“After a mean observation period of 53·9 months, 191 implants in 57 patients with OLP showed a survival rate of 95·3% (SD ± 21.2). No strict contraindication for the placement of implants seems to be justified in patients with OLP (…). Implant survival rates are comparable to those of patients without oral mucosal diseases.”
Guobis et al. [33]	2016	Systematic Review	3 case-control studies (1 retrospective, 1 prospective, 1 cross-sectional) with 106 patients and 278 implants	“Success of implant rehabilitation among treated OLP patients does not differ from the success rate in the general population. Implant survival and success rate was 100% vs. 96.8% in the control group.”
Strietzel et al. [34]	2019	Systematic review	9 studies (4 case reports, 1 case-control retrospective study, 1 case-control cross-sectional study, 1 case-control prospective study, 2 cohort retrospective studies with 100 patients and 302 implants	”After a mean follow-ip period of 44.6 months, a weighed mean values of implant survival rate of 98.3% was calculated (…) for patients with OLP (100 patients with 302 implants). Implant survival rates of patients affected are comparable to those of healthy patients.”
Xiong et al. [35]	2020	Systematic review and meta-analysis	2 studies (1 case-control prospective study and 1 case-control cross-sectional study) with 68 participants receiving 222 implants.	“Proportions of implants with peri-implant diseases (PIDs) between OLP and non-OLP groups were as follows: 19.6% (22/112) vs. 22.7% (25/110) for peri-implant mucositis and 17.0% (19/112) vs. 10.9% (12/110) for peri-implantitis. The meta-analysis revealed no recognizable difference in number of implants with PIDs (…) between OLP and non-OLP groups. Existing evidence does not support OLP as a suspected risk for peri-implant diseases.”
Torrejon-Moya et al. [36]	2020	Systematic review and meta-analysis	15 studies (10 case reports, 1 case-control retrospective study, 1 case-control cross-sectional study, 1 case-control prospective study, 1 cohort retrospective study, 1 cohort prospective study) with 110 patients. 3 studies included in meta-analysis (48 patients with OLP and 49 patients without OLP)	“According to the results of the meta-analysis, with a total sample of 48 patients with OLP and 49 patients without OLP, an odds ratio of 2.48 (95% CI 0.34–18.1) was established, with an I^2^ value of 0%. According to the Strength of Recommendation Taxonomy (SORT) criteria, level A can be established to conclude that patients with OLP can be rehabilitated with dental implants.”
Esimekara et al. [37]	2022	Systematic critical review	11 studies (5 case reports, 1 case-control retrospective study, 1 case-control cross sectional study, 1 case-control prospective study, 2 cohort retrospective studies, 1 cohort prospective study)	“This review suggested that dental implants may be considered as a safe and viable therapeutic option in the management of edentulous patients suffering from autoimmune diseases. (…) Results showed that dental implant survival rates were comparable to those reported in the general population. However, patients with (…) erosive OLP were more susceptible to developing peri-mucositis and increased marginal bone loss.”

## Data Availability

Not applicable.

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
