# Peer review of "Dental Implant Treatment in Patients Suffering from Oral Lichen Planus: A Narrative Review"

_ijerph, 2022, doi:10.3390/ijerph19148397_

Round 1
Reviewer 1 Report
Dear authors,
The review entitled "Dental implant treatment in patients suffering from oral lichen planus: a narrative review." presents an interesting topic about treatment with dental implants and the presence of oral lichen planus during this specific treatment. The review applied the search methodology for systematic reviews (PICO) and present a good collection of articles about this subject. However, the authors showed results from clinical studies and also, systematic reviews and meta-analyses. There is one methodological fault in this manuscript. The authors need to correct it and decide about a narrative review or systematic review.
1- If the authors decided to apply the PICO methodology, this manuscript can not show systematic reviews in your results. The studies presented in those systematic reviews should be presented in this current manuscript as a clinical studies one by one.
2- Systematic reviews and meta-analyses can not be considered part of the results. These studies can be only discussed as a discussion section or introduction.
3- If the authors would like to show all types of studies about this topic including systematic reviews, and meta-analyses, among others. The methodology for article search should not be PICO or a specific search for systemic review. Therefore, you should only use specific terms for search and include all types of articles as a narrative review.
4- Discussion section: Line 224- 229: There is no reference that supports the discussion that titanium is better recommended for this type of treatment. Moreover, the literature has been showing that prosthetic components in zirconia promote better conditions for the transmucosal region and higher biocompatibility. And this should be discussed.
DOI: 10.3390/ma14112825
DOI: 10.1097/ID.0000000000000167
DOi: 10.1016/j.dental.2019.04.010
5- The conclusion section should present that the study has some limitations. It is a narrative review, thus the authors should expose this. With the limitations of this narrative review... From the compilation of information found in the literature... Or another way, in order to show the conclusions are not new and yes, a summary of the literature.
Author Response
Response to Reviewer 1 Comments
Dear Reviewer,
first of all, thank You very much for Your insightful comments and Your effort to review our research.
Every objection that was pointed we addressed in the manuscript if possible (please see the attached Word file):
Point 1: The review applied the search methodology for systematic reviews (PICO) and present a good collection of articles about this subject. However, the authors showed results from clinical studies and also, systematic reviews and meta-analyses. There is one methodological fault in this manuscript. The authors need to correct it and decide about a narrative review or systematic review.
If the authors decided to apply the PICO methodology, this manuscript can not show systematic reviews in your results. The studies presented in those systematic reviews should be presented in this current manuscript as a clinical studies one by one.
If the authors would like to show all types of studies about this topic including systematic reviews, and meta-analyses, among others. The methodology for article search should not be PICO or a specific search for systemic review. Therefore, you should only use specific terms for search and include all types of articles as a narrative review.
Response 1: The aim of this manuscript was to present up-dated knowledge about recommendations, complications and survival rates of dental implants placed in patients suffering from oral lichen planus through a narrative review of the published literature. There were some methodological problems with the execution of this manuscript, that were misleading. Based on the Reviewer comments they were corrected. The search methodology was changed (PICO was crossed out, and only specific search terms were used), the results section was rewritten (the clinical studies were presented one by one) and tables were reorganized. Other parts of the manuscript (abstract, discussion, conclusions) were also changed. Some repetitions were removed through whole manuscript and language was corrected by a native speaker. Please check the improved manuscript.
To correct the manuscript the guidelines presented by Green BN, Johnson CD, Adams A. in „Writing narrative literature reviews for peer-reviewed journals: secrets of the trade. J Chiropratic Medicine 2006;5:101–117”were followed.
Point 2: Systematic reviews and meta-analyses cannot be considered part of the results. These studies can be only discussed as a discussion section or introduction.
Response 2: Systematic reviews and meta-analyses were moved from the Results section to the Discussion section. Table 2 was reorganized and also moved to the Discussion section. Please check the improved manuscript.
Point 3: Discussion section: Line 224- 229: There is no reference that supports the discussion that titanium is better recommended for this type of treatment. Moreover, the literature has been showing that prosthetic components in zirconia promote better conditions for the transmucosal region and higher biocompatibility. And this should be discussed.
DOI: 10.3390/ma14112825
DOI: 10.1097/ID.0000000000000167
DOI: 10.1016/j.dental.2019.04.010
Response 3: In this paragraph we reiterated recommendations done by Anitua et al. in their systematic review regarding prosthetic aspects of implant treatment in patients with oral lichen planus. However, the Reviewer is absolutely right about better biocompatibility of zirconia, and better tissue response to this material. Recommended articles were added to the discussion and reference list was updated. Please check the revised manuscript.
Point 4: The conclusion section should present that the study has some limitations. It is a narrative review, thus the authors should expose this. With the limitations of this narrative review... From the compilation of information found in the literature... Or another way, in order to show the conclusions are not new and yes, a summary of the literature.
Response 4: The inherent disadvantages of narrative review were described in the appropriate section of discussion and conclusion. Please check the revised manuscript.
Reviewer 2 Report
The review represent for clincians a very updated and useful paper. The thema is in very high interest because more and more patients are in a real need of implant supported prosthetises.
Appreciate the review and its contribution to further science.
Author Response
Response to Reviewer 2 Comments
Dear Reviewer,
first of all, thank You very much for Your effort to review our research.

Reviewer 3 Report
Dear Editor,
Regarding the submitted manuscript “ Dental implant treatment in patients suffering from oral lichen planus: a narrative review.” the presented study is intended to be a narrative review regarding dental implants placement in patients with oral lichen planus.
Although the authors mention in the title narrative review the presented work corresponds to a mixture between narrative and systematic review.
ROBIS Detailed appreciation
Phase 1: Assessing relevance
Although relevant the proposed work does not add anything novel to the already published literature.
Phase 2: Identifying concerns with the review process
DOMAIN 1: STUDY ELIGIBILITY CRITERIA
Describe the study eligibility criteria, any restrictions on eligibility and whether there was evidence that objectives and eligibility criteria were pre-specified.
While addressing the stated points in the ROBIS tools the concerns regarding specification of study eligibility criteria of this referee are:
Although not mandatory, the study protocol should be registered (ex: Prospero) since enables the readers to check for additional details and also to confirm that the study design was made à priori from the obtained results.
The eligibility criteria are not clear.
days/Months/Year in the time limits instead of just years (point 6 prisma)
The search should be performed without language restriction (point 6 prisma)
Time of follow up for elegibility
Grey literature search needs to be performed and the authors of the included studies contacted (point 7 prisma)
DOMAIN 2: IDENTIFICATION AND SELECTION OF STUDIES
Regarding the study itself it is advisable redoing the search adding the following criteria to prevent the selection bias:
- Contacting all the authors of the included studies to request for additional published or unpublished data
- Study appraisal calibration and evaluation need to be performed (kappa between reviewers in the critical appraisal)- the study selection and appraisal needs to be at least performed by two different authors
- Study selection according to prisma figure needs to be presented and methods adequately reported to allow to perform the search and obtain the same results
DOMAIN 3: DATA COLLECTION AND STUDY APPRAISAL
Risk of bias in the individual studies needs to be performed according to specified methods and this data used in the data synthesis (point 12 of prisma) and kappa reports between evaluators needs to be presented.
Taking a look on the obtained data we are looking to low level articles with small sample sizes without adding anything novel to the previously published systematic reviews.
DOMAIN 4: SYNTHESIS AND FINDINGS
Although with good intentions and with a good amount of performed work , the retrieved information is with low quality studies not allowing in this reviewer opinion for a publishable study.
Author Response
Response to Reviewer 3 Comments
Dear Reviewer,
first of all, thank You very much for Your insightful comments and Your effort to review our research.
Every objection that was pointed we addressed in the manuscript if possible (please see the attached Word file):
Point 1: Although the authors mention in the title narrative review the presented work corresponds to a mixture between narrative and systematic review.
Response 1: The aim of this manuscript was to present up-dated knowledge about recommendations, complications and survival rates of dental implants placed in patients suffering from oral lichen planus through a narrative review of the published literature. There were some methodological problems with the execution of this manuscript, that were misleading. Based on the Reviewer comments they were corrected. The search methodology was changed (PICO was crossed out, and only specific search terms were used), the results section was rewritten (the clinical studies were presented one by one) and tables were reorganized. Other parts of the manuscript (abstract, discussion, conclusions) were also changed. Some repetitions were removed through whole manuscript and language was corrected by a native speaker. Please check the improved manuscript.
To correct the manuscript the guidelines presented by Green BN, Johnson CD, Adams A. in „Writing narrative literature reviews for peer-reviewed journals: secrets of the trade. J Chiropratic Medicine 2006;5:101–117”were followed.
Point 2: ROBIS Detailed appreciation
Phase 1: Assessing relevance
Although relevant the proposed work does not add anything novel to the already published literature.
Phase 2: Identifying concerns with the review process
DOMAIN 1: STUDY ELIGIBILITY CRITERIA
Describe the study eligibility criteria, any restrictions on eligibility and whether there was evidence that objectives and eligibility criteria were pre-specified.
While addressing the stated points in the ROBIS tools the concerns regarding specification of study eligibility criteria of this referee are:
Although not mandatory, the study protocol should be registered (ex: Prospero) since enables the readers to check for additional details and also to confirm that the study design was made à priori from the obtained results.
The eligibility criteria are not clear.
days/Months/Year in the time limits instead of just years (point 6 prisma)
The search should be performed without language restriction (point 6 prisma)
Time of follow up for elegibility
Grey literature search needs to be performed and the authors of the included studies contacted (point 7 prisma)
DOMAIN 2: IDENTIFICATION AND SELECTION OF STUDIES
Regarding the study itself it is advisable redoing the search adding the following criteria to prevent the selection bias:
- Contacting all the authors of the included studies to request for additional published or unpublished data
- Study appraisal calibration and evaluation need to be performed (kappa between reviewers in the critical appraisal)- the study selection and appraisal needs to be at least performed by two different authors
- Study selection according to prisma figure needs to be presented and methods adequately reported to allow to perform the search and obtain the same results
DOMAIN 3: DATA COLLECTION AND STUDY APPRAISAL
Risk of bias in the individual studies needs to be performed according to specified methods and this data used in the data synthesis (point 12 of prisma) and kappa reports between evaluators needs to be presented.
Taking a look on the obtained data we are looking to low level articles with small sample sizes without adding anything novel to the previously published systematic reviews.
DOMAIN 4: SYNTHESIS AND FINDINGS
Although with good intentions and with a good amount of performed work , the retrieved information is with low quality studies not allowing in this reviewer opinion for a publishable study.
Response 2: Next points made by the Reviewer are based on ROBIS, which is used to evaluate systematic reviews. For narrative reviews such strict recommendations do not exist. The guidelines for narrative review preparation were presented by Green BN, Johnson CD, Adams A. in „Writting narrative literature reviews for peer-reviewed journals: secrets of the trade. J Chiropratic Medicine 2006;5:101–117” and we followed them while revising our manuscript. In the revised manuscript we crossed out all information that were misleading with respect to the type of manuscript. Search strategy was rewritten, together with inclusion and exclusion critearia. In the Results section only original articles were presented. We would like to reiterate that the aim of this manuscript was to provide up-dated information on the topic of dental implant treatment of patients with oral lichen planus, regarding its rationale, patients preparation, prosthetic considerations, flare-ups and their treatments, and possible complications through a narrative review. Thus some issues adressed by the Reviewer cannot be dealt with. We know that a lot of high-quality systematic reviews (and meta-analyses) on this topic exist, but they are not easily accessible for general practioners who may deal with patient suffering from oral lichen planus on daily basis. We wanted to bring practitioners up to date with accepted clinical protocols using readable format and broad perspective on a topic. We agree with the Reviewer that narrative reviews are one of the weakest forms of evidence to use for making clinical decisions in regard to patient care, which is why we prepared Table 2 with all existed systematic -reviews and their main conclusions. We described the inherent flaws of narrative reviews in appropriate section of the Discussion. We hope that this explanation will be enough for the Reviewer to allow for the publication of this narrative review.

Reviewer 4 Report
The author presents a narrative review describing implant treatment outcomes in patients with oral lichen planus. The author uses tables to describe the included studies' characteristics and makes suggestions for treating patients with OLP.
Lines 10 and 75: I suggest changing from “evaluate” to “describe” since this is a review paper.
Materials and methods section: the author refers to a PICO question and inclusion/exclusion criteria, which is a methodology of a systematic review. However, the manuscript is presented as a narrative review. Why this decision? Maybe you can consider structuring it as a scoping review.
The provided search formula is incomplete, so relevant articles may be missing.
Narrative reviews on this topic also exist. So why did the author not refer to them on the inclusion and exclusion criteria? And why they were not included?
Line 88: I believe the author is referring to RCT, which stands for randomized controlled trial and not randomized clinical trial.
Line 100: the author refers to the exclusion of in vitro studies, but this is not referred to in the exclusion criteria. Please correct.
Table 1: I suggest including the reasons for implant failure when described in the papers. Also, in studies with controls (for instance, Hérnandes et al., Czernin et al.…), I suggest adding the implant survival rates of the controls to facilitate the comparison.
Table 1: I suggest organizing this table, grouping the same study types.
Table 2: I suggest incrementing this table, adding the information on how many studies each SR included, how many patients and the studies type.
Some typing corrections are needed. See, for instance, lines 188 or 191.
Plagiarism software detected minor problems with previously published articles.
Author Response
Response to Reviewer 4 Comments
Dear Reviewer,
first of all, thank You very much for Your insightful comments and Your effort to review our research.
Every objection that was pointed we addressed in the manuscript if possible (please see the attached Word file):
Point 1: Lines 10 and 75: I suggest changing from “evaluate” to “describe” since this is a review paper.
Response 1: Checked and corrected.
Point 2: Materials and methods section: the author refers to a PICO question and inclusion/exclusion criteria, which is a methodology of a systematic review. However, the manuscript is presented as a narrative review. Why this decision? Maybe you can consider structuring it as a scoping review.
Response 2: The aim of this manuscript was to present up-dated knowledge about recommendations, complications and survival rates of dental implants placed in patients suffering from oral lichen planus through a narrative review of the published literature. There were some methodological problems with the execution of this manuscript, that were misleading. Based on the Reviewer comments they were corrected. The search methodology was changed (PICO was crossed out, and only specific search terms were used), the results section was rewritten (the clinical studies were presented one by one) and tables were reorganized. Please check the revised version of the manuscript.
To correct the manuscript the guidelines presented by Green BN, Johnson CD, Adams A. in „Writing narrative literature reviews for peer-reviewed journals: secrets of the trade. J Chiropratic Medicine 2006;5:101–117”were followed.
Point 3: The provided search formula is incomplete, so relevant articles may be missing.
Response 3: The search formula was expanded with new terms. Please check the revised manuscript.
Point 4: Narrative reviews on this topic also exist. So why did the author not refer to them on the inclusion and exclusion criteria? And why they were not included? Line 100: the author refers to the exclusion of in vitro studies, but this is not referred to in the exclusion criteria. Please correct.
Response 4: The inclusion and exclusion criteria were rewritten and mistakes were corrected. In the Results section appropriate studies were described one by one. In the Discussion section general remarks were made and already existing systematic reviews and meta-analyses were mentioned. Because as many as nine high-quality systematic reviews were found, we decided to focus on them.
Point 5: Line 88: I believe the author is referring to RCT, which stands for randomized controlled trial and not randomized clinical trial.
Response 5: Checked and corrected.
Point 6: Table 1: I suggest including the reasons for implant failure when described in the papers. Also, in studies with controls (for instance, Hérnandes et al., Czernin et al.…), I suggest adding the implant survival rates of the controls to facilitate the comparison. Table 1: I suggest organizing this table, grouping the same study types.
Response 6: The reasons for implant failure when described in the table (provided they were given in the original studies). The implant survival rates for controls were added. Table 1 was reorganized by grouping the same study types. Please check revised Table 1.
Point 7: Table 2: I suggest incrementing this table, adding the information on how many studies each SR included, how many patients and the studies type.
Response 7: Table 2 was rewritten and additional column with data regarding the information on number and type of included studies, number of analyzed patients and implants. Please check revised Table 2.
Point 8: Some typing corrections are needed. See, for instance, lines 188 or 191.
Response 8: Checked and corrected.

Round 2
Reviewer 1 Report
The author corrected all the points addressed with perfection.
Now, the manuscript presents a narrative review with a search methodology and systematic reviews supporting the results as a table in the discussion section.
Moreover, discussion was added and limitations of this review were addressed in the entire manuscript.
According to these corrections, this manuscript currently has merit for publication.
Author Response
Response to Reviewer 1 Comments
Dear Reviewer,
I would like to thank You very much for Your effort in reviewing our manuscript.

Reviewer 3 Report
It is this reviewers opinion that narrative reviews in scientific journals can be misleading regarding the available evidence and for this purpose the opinion to reject the paper continues the same
Author Response
Response to Reviewer 3 Comments
Dear Reviewer,
I understand your opinion and would like to thank You very much for Your time allocating in reviewing our manuscript.

Reviewer 4 Report
The author presents a narrative review describing implant treatment outcomes in patients with oral lichen planus. The performed modifications improved the manuscript quality. However, some clarifications are still needed.
Lines 80-81: The authors refer that the following terms were used in the literature search: “oral lichen planus” and “dental implant” or “oral implant” or “dental implant survival” or “dental implant complications”. The terms are incomplete and repeated. The plural is missing, and “dental implant survival” and “dental implant complications” are not necessary since searching for “dental implant” will retrieve the same results. Again, this is a problem because the literature search may be incomplete. Even if that is not the case, and since this is a narrative review, I suggest removing the description of the methods since this part is not mandatory in this kind of review.
Lines 81-82: the author states “All clinical reports, case series, prospective and retrospective studies on dental implant treatment in humans with oral lichen planus until May 2022 were included.” What do you mean by clinical reports? Case reports? RCTs? This section needs to be improved or removed, as I suggested above. You need to refer to study types (RCT, cohort, case-control…) and not prospective or retrospective.
Section 2: table 2 reports systematic reviews results, but the inclusion of systematic reviews is not reported in the materials and methods section. Also, there is no reference to the inclusion/exclusion of narrative reviews.
Table 2: please refer to study types (RCT, cohort, case-control…) and not just prospective or retrospective.
Author Response
Dear Reviewer,
first of all, thank You very much for Your insightful comments and Your effort to review our research.
Every suggestion that you made we addressed and changed the manuscript accordingly (please see the attached Word file):
Point 1: Lines 80-81: The authors refer that the following terms were used in the literature search: “oral lichen planus” and “dental implant” or “oral implant” or “dental implant survival” or “dental implant complications”. The terms are incomplete and repeated. The plural is missing, and “dental implant survival” and “dental implant complications” are not necessary since searching for “dental implant” will retrieve the same results. Again, this is a problem because the literature search may be incomplete. Even if that is not the case, and since this is a narrative review, I suggest removing the description of the methods since this part is not mandatory in this kind of review.
Response 1: This part was removed from the description of the methods.
Point 2: Lines 81-82: the author states “All clinical reports, case series, prospective and retrospective studies on dental implant treatment in humans with oral lichen planus until May 2022 were included.” What do you mean by clinical reports? Case reports? RCTs? This section needs to be improved or removed, as I suggested above. You need to refer to study types (RCT, cohort, case-control…) and not prospective or retrospective.
Response 2: Term “clinical reports” was changed to “case reports”. Study types were referred.
Point 3: Section 2: table 2 reports systematic reviews results, but the inclusion of systematic reviews is not reported in the materials and methods section. Also, there is no reference to the inclusion/exclusion of narrative reviews.
Response 3: Checked and corrected.
Point 4: Table 2: please refer to study types (RCT, cohort, case-control…) and not just prospective or retrospective.
Response 4: Checked and corrected.
